# Fast Finite-Time Stability and Its Application in Adaptive Control of High-Order Stochastic Nonlinear Systems

Yixuan Yuan and Junsheng Zhao *

School of Mathematics Science, Liaocheng University, Liaocheng 252000, China
* Correspondence: zhaojunshshao@163.com

**Abstract:** In this article, a new design method for an adaptive fast finite-time controller (FTC) is proposed for the finite-time stability (FTS) issue of a class of high-order stochastic nonlinear systems (HOSNSs) with unknown parameters. Using a power integrator technology and Lyapunov function approach, an adaptive state feedback controller is derived to ensure fast FTS of HOSNSs. The developed adaptive fast FTC is equipped with less settling time to obtain better steady-state accuracy compared with the traditional FTC. The effectiveness of the proposed adaptive control scheme is demonstrated by a numerical example.

**Keywords:** fast finite-time stability; adaptive control; high-order stochastic nonlinear systems; state feedback

## 1. Introduction

As typical dynamic systems, the controller design problem of high-order nonlinear systems (HONSs) has drawn great attention due to its important applications in the industry. However, HONSs are neither feedback linearization at the origin nor affine in the control input, which makes the control issues challenging. Fortunately, with the aid of the homogeneous domination technique [1] and by adding a power integrator technique [2], tremendous progress has been made for stability issues of HONSs [3–7]. Recently, the research on this subject has been further expended to HONSs with unknown nonlinear functions. To overcome the obstacles arising from the unknown nonlinearities, the adaptive control scheme is one of the most effective methods to deal with unknown functions; for instance, refer to [8–13] and references therein.

It should be noted that all the above results are asymptotically convergent, which cannot meet the transient performance requirements of practical application systems. As a matter of fact, it is expected that nonlinear system trajectories can converge to a Lyapunov stable equilibrium state in finite time, while also having better robustness and faster convergence speed. Based on this problem, Bhat et al. first presented the concept of FTS for continuous autonomous systems [14]. Moulay et al. extended the FTS theory of autonomous systems to non-autonomous systems [15]. In [16], a finite-time tracking control method of an uncertain nonlinear system was discussed in virtue of the approximation ability of fuzzy logic systems. Furthermore, there are also other remarkable achievements on FTS, such as [17–19] and the references therein.

Unfortunately, random disturbances are inevitable and often lead to the instability of the system in engineering systems. Consequently, it is very important to establish a stochastic system model and solve the issue of stability for stochastic nonlinear systems [20–27]. As far as the stability of the HOSNSs is investigated, the finite-time control is of great research value. For this purpose, Chen et al. and Yin et al. presented the finite-time attractiveness (FTA) concept and FTS theorem for stochastic nonlinear systems in [28,29] separately. Recently, by relaxing the constraint of the differential operator, Yu et al. proved that an unstable deterministic system can obtain finite-time convergence by Brownian noise [30], and a number of interesting results have been obtained [31–39].

With more in-depth research on FTS, researchers found that the convergence rate was even slower than the exponential when the initial state was far from the point of the origin. To handle this faultiness, Shen et al. made an important step to improve the convergence rate by adding a linear term into controller design processes [40]. Sun et al. presented fast FTS theory to improve the traditional finite-time control scheme [41]. To date, the so-called fast finite-time control scheme has received increasing interest from a variety of research communities, such as [42–44]. However, it should be pointed out that very few results have been available on fast FTS for stochastic nonlinear systems, and this leaves a gap that will be narrowed through our endeavors in this article.

Fully taking into consideration the practical application system requirements, this article strives to solve the adaptive fast finite-time control design issue for HOSNSs based on the abovementioned discussion. The main difficulties stem from the following. In comparison with the deterministic case, the stochastic nonlinear systems will generate more tedious inequalities since Itô stochastic differentiation involves not only gradients but also higher-order Hessian terms in the Lyapunov design process of HOSNSs. To solve this problem, we extend the fast FTS theory to the corresponding stochastic cases. The novelties of this article are summarized as follows:

1. Fast finite-time adaptive control strategy of deterministic systems is extended to stochastic cases, which provides a new idea for the FTS of HOSNSs, and expands its application scope in practical engineering successfully.
2. By choosing an appropriate Lyapunov function, an adaptive state feedback controller is constructed accordingly to ensure that equilibrium at the origin of the closed loop systems is fast finite-time stable in probability.
3. Compared with previous studies, it is shown that the fast finite-time adaptive control strategy not only improves the control speed significantly, but also reduces the settling time effectively.

*Notations:* To process further, we introduce the following useful notations. $\mathbb{R}^m$ is the Euclidean space with dimension $m$ for $m = 1, 2, \cdots, n$ and $\mathbb{R}^{\geq 1}_{odd} = \{\frac{a}{b} | a, b$ are positive odd integers satisfying $a \geq b\}$. $V \in \mathcal{C}^2$ means that $V$ is a second-order differentiable continuous function and its second derivative is also continuous. Consider stochastic nonlinear system $dx = f(t, x)dt + g(t, x)d\omega$; the differential operator $\mathcal{L}$ is defined as $\mathcal{L}V(x, t) = \frac{\partial V}{\partial t} + \frac{\partial V}{\partial x}f(x, t) + \frac{1}{2}trace[g(x, t)^T \frac{\partial^2 V}{\partial x^2} g(x, t)]$. For all $q \in \mathbb{R}^+$ and $y \in \mathbb{R}$, define $\lceil y \rceil^q = |y|^q sign(y)$.

## 2. Preliminaries

Now, a class of stochastic nonlinear systems are considered as follows:

$$
\begin{cases}
dx_1 = x_2^{p_1} dt + f_1(\bar{x}_2, d)dt + g_1^T(x_1, d)d\omega, \\
dx_2 = x_3^{p_2} dt + f_2(\bar{x}_3, d)dt + g_2^T(\bar{x}_2, d)d\omega, \\
\quad \vdots \\
dx_n = u^{p_n} dt + f_n(x, u, d)dt + g_n^T(x, d)d\omega, \\
x(0) = (x_{10}, \cdots, x_{n0})^T,
\end{cases}
\tag{1}
$$

where $x(t) = (x_1, x_2, \cdots, x_n)^T \in \mathbb{R}^n$ is the system state, $u(t) \in \mathbb{R}$ is the control input, $d \in \mathbb{R}^r$ is a parameter vector denoting an unknown value, and $p_i$ satisfies $p_i \in \mathbb{R}^{\geq 1}_{odd}$. The functions $f_i(\cdot)$ and $g_i(\cdot)$ are continuous with $f_i(0, d) = 0$ and $g_i(0, d) = 0$ for $i = 1, 2, \cdots, n$. $\omega$ is a standard Brownian motion defined on a probability space $(\Omega, \mathcal{F}, \mathcal{P})$. The following assumptions are needed in this paper.

**Assumption 1.** *There exist known nonnegative smooth functions $\bar{f}_i$ and $\bar{g}_i$ such that, for arbitrary $i = 1, \cdots, n$*

$$|f_i(\bar{x}_{i+1}, d)| \leq \kappa_i |x_{i+1}|^{p_i} + \theta \sum_{l=1}^{i} |x_l|^{\mu_{i,l} + \frac{r_i + \omega}{r_l}} \bar{f}_i(\bar{x}_i), \tag{2}$$

$$||g_i(\bar{x}_{i+1}, d)|| \leq \theta \sum_{l=1}^{i} |x_l|^{\bar{\mu}_{i,l} + \frac{2r_i + \omega}{2r_l}} \bar{g}_i(\bar{x}_i), \tag{3}$$

*where $0 \leq \kappa_i < 1$, $\omega \in (-\frac{1}{\sum_{l=1}^{n} 2p_0 \cdots p_{l-1}}, 0) \leq 0$ with $p_0 = 1$, and $\mu_{i,l} \geq 0$, $\bar{\mu}_{i,l} \geq 0$, $r_1 = \frac{1}{2}$, $r_{i+1} = \frac{r_i + \omega}{p_i}$ are constants.*

The following definitions and lemmas are needed for the processes of controller derivation.

**Definition 1.** [29] *Consider a stochastic nonlinear system*

$$dx(t) = f(t, x(t))dt + g(t, x(t))dw(t), \tag{4}$$

*where $x(0) = x_0 \in \mathbb{R}^n$ is the initial value. For any $x_0 \in \mathbb{R}$, if the solution $x(t; x_0)$ satisfies the following conditions, then the system (4) is finite-time stable in probability.*

1. *Finite-time attractiveness in probability: The stochastic settling time $\tau_{x_0}^* = \inf\{t : x(t; x_0) = 0\}$ with initial value $x_0 \in \mathbb{R}^n \setminus \{0\}$ is finite almost surely; in other words, $P\{\tau_{x_0} < +\infty = 1\}$;*
2. *Stability in probability: For $\epsilon \in (0,1)$ and $s > 0$, which means that there exists a $\sigma = \sigma(\epsilon, s) > 0$ such that $P\{|x(t; x_0)| < s, \forall t \geq 0\} \geq 1 - \epsilon$, whenever $|x_0| < \sigma$.*

**Lemma 1.** [42]
*For $v_1 \in \mathbb{R}^m$, $v_2 \in \mathbb{R}^n$, if $\aleph(v_1, v_2)$ is a continuous function, then*

$$|\aleph(v_1, v_2)| \leq a(v_1) + b(v_2),$$
$$|\aleph(v_1, v_2)| \leq c(v_1)d(v_2).$$

*where $a(v_1) \geq 0$, $b(v_2) \geq 0$, $c(v_1) \geq 1$, $d(v_2) \geq 1$ are smooth functions.*

**Lemma 2.** [41] *Let real numbers $v_1, v_2 > 0$ and a smooth function $\varepsilon(a, b) > 0$; then, for arbitrary $a \in \mathbb{R}$, $b \in \mathbb{R}$*

$$|a|^{v_1}|b|^{v_2} \leq \frac{v_1}{v_1 + v_2}\varepsilon(\cdot)|a|^{v_1 + v_2} + \frac{v_2}{v_1 + v_2}\varepsilon^{-\frac{v_1}{v_2}}(\cdot)|v_2|^{v_1 + v_2}.$$

**Lemma 3.** [35] *For a given $p = \frac{a}{b} \in \mathbb{R}_{odd}^{\geq 1}$, $b \geq 1$, then for $v_1, v_2 \in \mathbb{R}$,*

$$|v_1 + v_2|^{\frac{1}{p}} \leq |v_1|^{\frac{1}{p}} + |v_2|^{\frac{1}{p}},$$
$$|v_1 + v_2|^p \leq 2^{p-1}(|v_1|^p + |v_2|^p),$$
$$|v_1^{\frac{1}{p}} - v_2^{\frac{1}{p}}| \leq 2^{\frac{p-1}{p}}|v_1 - v_2|^{\frac{1}{p}},$$
$$|v_1^{\frac{a}{b}} - v_2^{\frac{a}{b}}| \leq 2^{\frac{b-1}{b}}|\lceil v_1 \rceil^a - \lceil v_2 \rceil^a|^{\frac{1}{b}}.$$

**Lemma 4.** [41] *For $b \geq 1$, $v \in \mathbb{R}$, if $\hbar(v) = sign(v)|v|^b$ is $\mathcal{C}^1$, then*

$$\dot{\hbar}(v) = b|v|^{b-1}.$$

**Lemma 5.** [30] *For stochastic systems (4), if existing, $\mathcal{C}^2$ function $V : \mathbb{R}^n \to \mathbb{R}_+$ and constants $c > 0$, $\gamma \in (0,1)$, the following inequality arises:*

$$\mathcal{L}V(x) \leq 0, \quad \forall x \in \mathbb{R}^n,$$

$$K(V(x))[ck(V(x)) + \mathcal{L}V(x)] \leq \frac{K'(V(x))}{2} \mid \frac{\partial V}{\partial x} g(x) \mid^2, \quad \forall x \in \mathbb{R}^n \backslash \{0\},$$

*where $K(V(x)) = V^\gamma(x) + V(x)$ is a continuous differentiable function with the derivative $K'(V(x)) \geq 0$ and $K(V(x)) > 0$ for any $V(x) > 0$ and*

$$\int_0^\epsilon \frac{1}{K(V(x))} ds < \infty, \quad \forall \epsilon > 0,$$

*then, the system (4) is finite-time stable in probability, and the settling time satisfies*

$$E\tau_{x_0} \leq \frac{1}{c} \int_0^{V(x_0)} \frac{1}{K(V(x))} ds.$$

**Lemma 6.** [39] *For autonomous system (4), if existing, nonnegative and radially unbounded function $V \in \mathcal{C}^2$, and there holds for all $x \in \mathbb{R}^n$*

$$\mathcal{L}V(x) \leq 0.$$

*This means that autonomous system (4) has a solution for all initial data.*

### 3. Design Procedures

We now derive the state feedback controller. Firstly, the following transformation formations are introduced:

$$\begin{cases} \xi_1 = \lceil x_1 \rceil^{\frac{1}{r_1}}, \\ \xi_k = \lceil x_k \rceil^{\frac{1}{r_k}} - \lceil x_k^*(\bar{x}_{k-1}, \hat{\Theta}) \rceil^{\frac{1}{r_k}}, \\ x_k^*(\bar{x}_{k-1}, \hat{\Theta}) = -\lceil \xi_{k-1} \rceil^{r_k} \beta_{k-1}^{r_k}(\bar{x}_{k-1}(t), \hat{\Theta}), \\ u = x_{n+1}^*, \; k = 1, \cdots, n, \end{cases} \tag{5}$$

where $\beta_1, \cdots, \beta_k$ are $\mathcal{C}^2$ functions, $x_k^*(\bar{x}_{k-1}, \hat{\Theta})$ is called a virtual controller, and $\hat{\Theta}(t)$ is the estimate of the unknown constant parameter $\Theta = max\{\theta, \theta^2\}$. In view of (5), it follows that

$$\dot{\hat{\Theta}} = \sum_{i=1}^k \tau_i \xi_i^4,$$

$$u(t) = x_{n+1}^* = -\left(\frac{\gamma + \phi_n(\bar{x}_n, \hat{\Theta})}{1 - \kappa_n}\right)^{\frac{1}{p_n}} \lceil \xi_n \rceil^{r_{n+1}}. \tag{6}$$

Moreover, the current task is the confirmation of $\phi_n$ and $\tau_i$ in (6) to implement the detailed expression of controller $u(t)$. To this aim, one can consider $W_k : \mathbb{R}^k \to \mathbb{R}$ as

$$W_k(\bar{x}_k) = \int_{x_{k-1}^*}^{x_{k-1}} \lceil \lceil s \rceil^{\frac{1}{r_{k-1}}} - \lceil x_{k-1}^* \rceil^{\frac{1}{r_{k-1}}} \rceil^{4 - r_k p_{k-1}} ds. \tag{7}$$

*Step 1:* Let $V_1 = W_1 + \frac{1}{2}\tilde{\Theta}^2$, where $\tilde{\Theta}(t) = \Theta - \hat{\Theta}(t)$. Using (5) and Lemma 3, one has $|x_2|^{p_1} \leq |\xi_2|^{r_2 p_1} + |\xi_2^*|^{p_1}$. In addition, there holds

$$|\xi_1|^{-\omega} \leq (1 + \xi_1^2)^{-\frac{\omega}{2}}.$$

Then, by Assumption 1 and Lemma 1, one can determine that

$$|f_1| \leq \kappa_1 |x_2|^{p_1} + \tilde{\Theta}|\xi_1|^{r_1+\omega} l_{11} + \hat{\Theta}|\xi_1|^{r_1+\omega} \bar{l}_{11},$$
$$\|g_1^T g_1\| \leq \tilde{\Theta}|\xi_1|^{2r_1+\omega} l_{12} + \hat{\Theta}|\xi_1|^{2r_1+\omega} \bar{l}_{12},$$

where $\bar{l}_{11}(x_1) \leq l_{11} \triangleq |x_1|^{\mu_{11}} \bar{f}_1(x_1)$, $\bar{l}_{12}(x_1) \leq l_{12} \triangleq |x_1|^{2\bar{\mu}_{11}} \|\bar{g}_1^T \bar{g}_1\|$ are smooth and positive functions. It should be noted that

$$m_1 W_1 \leq m_1 \lceil \xi_1 \rceil^{4-r_2 p_1 + r_1} \leq m_1 \xi_1^4 (1 + \xi_1^2)^{-\frac{\omega}{2}}.$$

As a result, there holds

$$
\begin{aligned}
\mathcal{L}V_1 \leq &-(n-1+\gamma)\xi_1^4 - m_1 W_1 + \lceil \xi_1 \rceil^{4-r_2 p_1}(x_2^{p_1} - x_2^{*p_1}) \\
&+ \tilde{\Theta}(\tau_1 \xi_1^4 - \dot{\hat{\Theta}}) + \kappa_1 |\xi_1^{4-r_2 p_1} x_2^{p_1}| \\
&+ \lceil \xi_1 \rceil^{4-r_2 p_1}(x_2^{*p_1} + (n-1+\gamma+\hat{\Theta}\bar{l}_1 + \phi_1)\lceil \xi_1 \rceil^{r_2 p_1}),
\end{aligned}
\tag{8}
$$

where $\bar{l}_1(x_1) = \bar{l}_{11} + \frac{4-r_2 p_1}{2r_1}\bar{l}_{12}$, $\phi_1 = m_1(1 + \xi_1^2)^{-\frac{\omega}{2}}$, $\tau_1(x_1) = l_{11} + \frac{4-r_2 p_1}{2r_1} l_{12}$ are $\mathcal{C}^2$ functions and $m$ is a positive number. With the choice of

$$\beta_1(x_1, \hat{\Theta}) = \left(\frac{n-1+\gamma+\hat{\Theta}\bar{l}_1 + \phi_1}{1-\kappa_1}\right)^{\frac{1}{r_2 p_1}}, \tag{9}$$

it is directly deduced from (9) that

$$
\begin{aligned}
\mathcal{L}V_1 \leq &-(n-1+\gamma)\xi_1^4 + \lceil \xi_1 \rceil^{4-r_2 p_1}(x_2^{p_1} - x_2^{*p_1}) \\
&+ \tilde{\Theta}(\tau_1 \xi_1^4 - \dot{\hat{\Theta}}) + \kappa_1 |\xi_1|^{4-r_2 p_1}|\xi_2|^{r_2 p_1}.
\end{aligned}
\tag{10}
$$

*Step 2:* Let $V_2 = V_1 + W_2$ be a Lyapunov candidate function; it can be deduced from (10) that

$$
\begin{aligned}
\mathcal{L}V_2 \leq &-(n-1+\gamma)\xi_1^4 - m_1 W_1 - m_1 W_2 + \tilde{\Theta}(\tau_1 \xi_1^4 - \dot{\hat{\Theta}}) + \lceil \xi_2 \rceil^{4-r_3 p_2}(x_3^{p_2} - x_3^{*p_2}) \\
&+ \frac{\partial W_2}{\partial \hat{\Theta}}\dot{\hat{\Theta}} + \lceil \xi_1 \rceil^{4-r_2 p_1}(x_2^{p_1} - x_2^{*p_1}) + \kappa_1 |\xi_1|^{4-r_2 p_1}|\xi_2|^{r_2 p_1} + m_1 W_2 \\
&+ \frac{\partial W_2}{\partial x_2}(x_3^{*p_2} + f_2) + \frac{\partial W_2}{\partial x_1}(x_2^{p_1} + f_1) + \frac{1}{2}\sum_{i,j=1}^{2}\frac{\partial^2 W_2}{\partial x_i \partial x_j}\|g_i^T g_j\|.
\end{aligned}
\tag{11}
$$

In view of Lemmas 2 and 3, there exists a constant $\phi_{21} \geq 0$ such that

$$
\begin{aligned}
&\lceil \xi_1 \rceil^{4-r_2 p_1}(x_2^{p_1} - x_2^{*p_1}) + \kappa_1 |\xi_1|^{4-r_2 p_1}|\xi_2|^{r_2 p_1} \\
&\leq (2^{1-r_2 p_1} + \kappa_1)|\xi_1|^{4-r_2 p_1}|\xi_2|^{r_2 p_1} \\
&\leq \frac{1}{5}\xi_1^4 + \phi_{21}\xi_2^4.
\end{aligned}
\tag{12}
$$

By Assumption 1 and Lemmas 1–3, some tedious calculations show that

$$\frac{\partial W_2}{\partial x_2}(x_3^{*p_2} + f_2) \leq \lceil \xi_2 \rceil^{4-r_3 p_2} x_3^{*p_2} + \kappa_2 \lceil \xi_2 \rceil^{4-r_3 p_2}|x_3|^{p_2} + \tilde{\Theta}\tau_{21}\xi_2^4 + \frac{1}{5}\xi_1^4 + \phi_{22}\xi_2^4, \tag{13}$$

$$\frac{\partial W_2}{\partial x_1}(x_2^{p_1} + f_1) \leq \tilde{\Theta}\tau_{22}\xi_2^4 + \frac{1}{5}\xi_1^4 + \phi_{23}\xi_2^4, \tag{14}$$

$$\frac{1}{2}\sum_{i,j=1}^{2}\frac{\partial^2 W_2}{\partial x_i \partial x_j}\|g_i^T g_j\| \leq \tilde{\Theta}\tau_{23}\xi_2^4 + \frac{1}{5}\xi_1^4 + \phi_{24}\xi_2^4, \tag{15}$$

where $\tau_{2i}(\bar{x}_2, \hat{\Theta})$ are nonnegative continuous functions and $\phi_{2j} \geq 0$ are smooth functions for $i = 1, 2, 3$, $j = 2, 3, 4$. In addition, it is worth noting that

$$m_1 W_2 \leq 2^{1-r_2} m_1 (1 + \xi_2^2)^{-\frac{\omega}{2}} \xi_2^4 \triangleq \phi_{25} \xi_2^4, \tag{16}$$

$$\frac{\partial W_2}{\partial \hat{\Theta}} \sum_{i=1}^{2} \tau_i \xi_i^4 \leq \frac{1}{5} \xi_1^4 + \phi_{26} \xi_2^4, \tag{17}$$

where $\phi_{25} \geq 0$, $\phi_{26} \geq 0$, $\tau_2 = \sum_{i=1}^{3} \tau_{2i}$ are smooth functions. Let $\phi_2 = \sum_{i=1}^{6} \phi_{2i}$; substituting (12)–(17) into (11) and choosing

$$\beta_2(\bar{x}_2, \hat{\Theta}) = \left( \frac{n - 2 + \gamma + \phi_2}{1 - \kappa_2} \right)^{\frac{1}{r_3 p_2}},$$

one can obtain

$$\begin{aligned}
\mathcal{L}V_2 &\leq -(n - 2 + \gamma)(\xi_1^4 + \xi_2^4) - m_1(W_1 + W_2) + \lceil \xi_2 \rceil^{4 - r_3 p_2}(x_3^{p_2} - x_3^{*p_2}) \\
&\quad + (\tilde{\Theta} - \frac{\partial W_2}{\partial \hat{\Theta}})(\sum_{i=1}^{2} \tau_i \xi_i^4 - \dot{\hat{\Theta}}) + \kappa_2 |\xi_2|^{4 - r_3 p_2} |\xi_3|^{r_3 p_2}.
\end{aligned} \tag{18}$$

*Step k:* Assume that one can choose a $\mathcal{C}^2$ Lyapunov candidate function $V_{k-1} = V_{k-2} + W_{k-1}$ to guarantee

$$\begin{aligned}
\mathcal{L}V_{k-1} &\leq -(n + 1 - k + \gamma) \sum_{i=1}^{k-1} \xi_i^4 - m_1 \sum_{i=1}^{k-1} W_i + \lceil \xi_{k-1} \rceil^{4 - r_k p_{k-1}}(x_k^{p_{k-1}} - x_k^{*p_{k-1}}) \\
&\quad + (\tilde{\Theta} - \sum_{i=2}^{k-1} \frac{\partial W_i}{\partial \hat{\Theta}})(\sum_{i=1}^{k-1} \tau_i \xi_i^4 - \dot{\hat{\Theta}}) + \kappa_{k-1} |\xi_{k-1}|^{4 - r_k p_{k-1}} |\xi_k|^{r_k p_{k-1}}.
\end{aligned} \tag{19}$$

Consider a Lyapunov candidate function

$$V_k = V_{k-1} + W_k = V_{k-1} + \int_{x_k^*}^{x_k} \lceil \lceil s \rceil^{\frac{1}{r_k}} - \lceil x_k^* \rceil^{\frac{1}{r_k}} \rceil^{4 - r_{k+1} p_k} ds, \tag{20}$$

and it can be deduced from (19) and (20) that

$$\begin{aligned}
\mathcal{L}V_k &\leq -(n + 1 - k + \gamma) \sum_{i=1}^{k-1} \xi_i^4 - m_1 \sum_{i=1}^{k} W_i + \lceil \xi_{k-1} \rceil^{4 - r_k p_{k-1}}(x_k^{p_{k-1}} - x_k^{*p_{k-1}}) \\
&\quad + (\tilde{\Theta} - \sum_{i=2}^{k-1} \frac{\partial W_i}{\partial \hat{\Theta}})(\sum_{i=1}^{k-1} \tau_i \xi_i^4 - \dot{\hat{\Theta}}) + \kappa_{k-1} |\xi_{k-1}|^{4 - r_k p_{k-1}} |\xi_k|^{r_k p_{k-1}} + \frac{\partial W_k}{\partial \hat{\Theta}} \dot{\hat{\Theta}} \\
&\quad + m_1 W_k + \frac{\partial W_k}{\partial x_k}(x_{k+1}^{p_k} + f_k) + \sum_{i=1}^{k-1} \frac{\partial W_k}{\partial x_i}(x_{i+1}^{p_i} + f_i) \\
&\quad + \frac{1}{2} \left( \sum_{i,j=1}^{k-1} | \frac{\partial^2 W_k}{\partial x_i \partial x_j} | \, \|g_i^T g_j\| + 2 \sum_{i=1}^{k-1} | \frac{\partial^2 W_k}{\partial x_k \partial x_i} | \, \|g_k^T g_i\| + | \frac{\partial^2 W_k}{\partial x_k^2} | \, \|g_k^T g_k\| \right).
\end{aligned} \tag{21}$$

First, the application of Lemmas 2 and 3 leads to

$$\begin{aligned}
&\lceil \xi_{k-1} \rceil^{4 - r_k p_{k-1}}(x_k^{p_{k-1}} - x_k^{*p_{k-1}}) + \kappa_{k-1} |\xi_{k-1}|^{4 - r_k p_{k-1}} |\xi_k|^{r_k p_{k-1}} \\
&\leq (2^{1 - r_k p_{k-1}} + \kappa_{k-1}) |\xi_{k-1}|^{4 - r_k p_{k-1}} |\xi_k|^{r_k p_{k-1}} \\
&\leq \frac{1}{5} \xi_{k-1}^4 + \phi_{k1} \xi_k^4,
\end{aligned} \tag{22}$$

where $\phi_{k1}$ is a positive constant.

Second, it can be deduced from Assumption 1 and Lemmas 1–3 and some tedious calculations that

$$\sum_{i=1}^{k-1} \frac{\partial W_k}{\partial x_i}(x_{i+1}^{p_i} + f_i) \leq \tilde{\Theta}\tau_{k1}\xi_k^4 + \frac{1}{5}\xi_{k-1}^4 + \frac{1}{4}\sum_{i=1}^{k-2}\xi_i^4 + \phi_{k2}\xi_k^4, \tag{23}$$

$$\frac{\partial W_k}{\partial x_k}(x_{k+1}^{p_k} + f_k) \leq \lceil \xi_k \rceil^{4-r_{k+1}p_k} x_{k+1}^{p_k} + \kappa_k |\xi_k|^{4-r_{k+1}p_k}|x_{k+1}|^{p_k} \tag{24}$$

$$+ \tilde{\Theta}\tau_{k2}\xi_k^4 + \frac{1}{5}\xi_{k-1}^4 + \frac{1}{4}\sum_{i=1}^{k-2}\xi_i^4 + \phi_{k3}\xi_k^4.$$

In addition, there holds

$$\frac{1}{2}\left(\sum_{i,j=1}^{k-1}\left|\frac{\partial^2 W_k}{\partial x_i \partial x_j}\right| \|g_i^T g_j\| + 2\sum_{i=1}^{k-1}\left|\frac{\partial^2 W_k}{\partial x_k \partial x_i}\right| \|g_k^T g_i\| + \left|\frac{\partial^2 W_k}{\partial x_k^2}\right| \|g_k^T g_k\|\right)$$

$$\leq C\left[\sum_{i,j=1,i\neq j}^{k-1}\left(\int_{x_k^*}^{x_k}\lceil\lceil s\rceil^{\frac{1}{r_k}} - \lceil x_k^*\rceil^{\frac{1}{r_k}}\rceil^{2-r_{k+1}p_k}ds \cdot \left|\frac{\partial(-\lceil x_k^*\rceil^{\frac{1}{r_k}})}{\partial x_i}\right| \left|\left|\frac{\partial(-\lceil x_k^*\rceil^{\frac{1}{r_k}})}{\partial x_j}\right|\right.\right.$$

$$+ \int_{x_k^*}^{x_k}\lceil\lceil s\rceil^{\frac{1}{r_k}} - \lceil x_k^*\rceil^{\frac{1}{r_k}}\rceil^{3-r_{k+1}p_k}ds \cdot \left|\frac{\partial^2(-\lceil x_k^*\rceil^{\frac{1}{r_k}})}{\partial x_i \partial x_j}\right|\right)\|g_i^T g_j\| \tag{25}$$

$$+ \sum_{i=1}^{k-1}|\xi_k|^{3-r_{k+1}p_k} \cdot \frac{\partial(-\lceil x_k^*\rceil^{\frac{1}{r_k}})}{\partial x_i}\|g_k^T g_i\| + |\xi_k|^{3-r_{k+1}p_k} \cdot |x_k|^{\frac{1}{r_k}-1}\|g_k^T g_k\|\right]$$

$$\leq \tilde{\Theta}\tau_{k3}\xi_k^4 + \frac{1}{5}\xi_{k-1}^4 + \frac{1}{4}\sum_{i=11}^{k-2}\xi_i^4 + \phi_{k4}\xi_k^4,$$

where $\tau_{ki}(\bar{x}_2, \hat{\Theta})$ and $\phi_{kj} \geq 0$ are nonnegative smooth functions for $i = 1, 2, 3$, $j = 2, 3, 4$.

Third, by Lemmas 1 and 3, there exist smooth functions $\phi_{k5} \geq 0$, $\phi_{k6} \geq 0$, $\tau_k = \sum_{i=1}^{3}\tau_{ki}$ such that

$$m_1 W_k \leq 2^{1-r_k}m_1(1 + \xi_k^2)^{-\frac{\omega}{2}}\xi_k^4 \triangleq \phi_{k5}\xi_k^4, \tag{26}$$

$$\frac{\partial W_k}{\partial \hat{\Theta}}\sum_{i=1}^{k-1}\tau_i\xi_i^4 + \sum_{i=2}^{k}\frac{\partial W_i}{\partial \hat{\Theta}}\tau_k\xi_k^4 \leq \frac{1}{5}\xi_{k-1}^4 + \frac{1}{4}\sum_{i=1}^{k-2}\xi_i^4 + \phi_{k6}\xi_k^4. \tag{27}$$

Now, take the virtual control signal $x_{k+1}^*$ as

$$x_{k+1}^*(\bar{x}_k, \hat{\Theta}) = -\beta_k^{r_k}(\bar{x}_k, \hat{\Theta})\lceil \xi_k \rceil^{r_{k+1}}$$
$$= -\left(\frac{n - k + \gamma + \phi_k}{1 - \kappa_k}\right)^{\frac{1}{p_k}}\lceil \xi_k \rceil^{r_{k+1}},$$

where $\phi_k = \sum_{i=1}^{6}\phi_{ki}$.

Bringing the above inequalities into (21) yields

$$\mathcal{L}V_k \leq -(n - k + \gamma)\sum_{i=1}^{k}\xi_i^4 - m_1\sum_{i=1}^{k}W_i + \lceil \xi_k \rceil^{4-r_{k+1}p_k}(x_{k+1}^{p_k} - x_{k+1}^{*p_k})$$

$$+ \left(\tilde{\Theta} - \sum_{i=2}^{k}\frac{\partial W_i}{\partial \hat{\Theta}}\right)\left(\sum_{i=1}^{k}\tau_i\xi_i^4 - \dot{\hat{\Theta}}\right) + \kappa_k|\xi_k|^{4-r_{k+1}p_k}|\xi_{k+1}|^{r_{k+1}p_k}. \tag{28}$$

*Step n:* Using the above inductive arguments, one can derive that

$$
\dot{\hat{\Theta}} = \sum_{i=1}^{k} \tau_i \xi_i^4,
$$

$$
u(t) = x_{n+1}^* = -\left(\frac{\gamma + \phi_n(\bar{x}_n, \hat{\Theta})}{1 - \kappa_n}\right)^{\frac{1}{p_n}} \lceil \xi_n \rceil^{r_{n+1}}
$$

$$
\triangleq -\beta_n^{r_{n+1}}(\bar{x}_n) \lceil \xi_n \rceil^{r_{n+1}},
$$

$(29)$

where $\beta_n \geq 0$ is a smooth function. Choose an integral Lyapunov function $V_n(\bar{x}_n, \Theta)$ such that

$$
\mathcal{L}V_n \leq -\gamma \sum_{i=1}^{n} \xi_i^4 - m_1 \sum_{i=1}^{n} W_i(\bar{x}_i) \leq -\gamma W(x)^{\frac{4}{4-\omega}} - m_1 W(x),
$$

$(30)$

where $W(x) = \sum_{i=1}^{n} W_i(\bar{x}_i)$.

## 4. Main Results

**Theorem 1.** *Under adaptive control law (29) and (30), stochastic system (1) and trajectory $(x(t), \hat{\Theta}(t))$ are bounded in probability and $x(t)$ is finite-time stable in probability for all initial states $(x(t), \hat{\Theta}(t))$.*

**Proof.** (i) To begin with, we prove that the closed-loop system has a solution.

Obviously, $V_1 = \int_0^{x_1} \lceil s_1 \rceil^{\frac{4-r_2 p_1}{r_1}} ds + \frac{1}{2}\tilde{\Theta}^2$ is positive definite and for $V_1(x) \to \infty$, $\|x\| \to \infty$. Supposing that $V_{k-1}$ is positive definite and radially unbounded with respect to $\bar{x}_{k-1}$ and $\tilde{\Theta}$, we now prove that $V_k$ is positive definite and radially unbounded with respect to $\bar{x}_k$ and $\tilde{\Theta}$. By Lemma 1, one has

$$
W_i = \int_{x_i^*}^{x_i} \lceil \lceil s_1 \rceil^{\frac{1}{r_i}} - \lceil x_i^* \rceil^{\frac{1}{r_i}} \rceil^{4-r_{i+1}p_i} ds \geq \frac{r_i}{2-\omega} 2^{\frac{(2-r_{i+1}p_i)(r_i-1)}{r_i}} |x_i - x_i^*|^{\frac{2-\omega}{r_i}}.
$$

$(31)$

Therefore, there exists a $c_i = \frac{r_i}{2-\omega} \times 2^{\frac{(2-r_{i+1}p_i)(r_i-1)}{r_i}}$, such that

$$
V_k(\bar{x}_k) = \sum_{i=1}^{k} W_i + \frac{1}{2}\tilde{\Theta}^2 \geq \sum_{i=1}^{k} c_i \mid x_i - x_i^* \mid^{\frac{2-\omega}{r_i}} + \frac{1}{2}\tilde{\Theta}^2
$$

$$
\geq \sum_{i=1}^{k} c_i \mid (\xi_i + \lceil x_i^* \rceil^{\frac{1}{r_i}})^{r_i} - x_i^* \mid^{\frac{2-\omega}{r_i}} + \frac{1}{2}\tilde{\Theta}^2,
$$

$(32)$

that is, $V_k$ is a positive definite function. In view of $\bar{x}_{k+1} = [\bar{x}_k, x_{k+1}]$, one has $\|\bar{x}_{k+1}\| \to \infty$, which means that $\|\bar{x}_k\| \to \infty$ or $x_{k+1} \to \infty$. On one hand, by $V_{k+1} \geq V_k$ and $\|\bar{x}_k\| \to \infty$, it is not difficult to derive $V_{k+1} \to \infty$, as $\|\bar{x}_{k+1}\| \to \infty$. On the other hand, according to (32), $x_{k+1} \to \infty$ and the continuity of $x_k^*$ ensures $V_{k+1} \to \infty$, as $\|\bar{x}_{k+1}\| \to \infty$. Consequently, $V_k(x, \tilde{\Theta}) \to \infty$, $\|x\| \to \infty$, and $V_k(x, \tilde{\Theta}) \to \infty$, $\tilde{\Theta} \to \infty$. For stochastic systems (1), it can be observed from (30) and Lemma 6 that there is a solution.

Based on the properties of positive definite $V_n$ and radially unbounded with respect to $\bar{x}_{k-1}$, $\tilde{\Theta}$, and (30), it follows that the result is bounded in the probability of $x$ and $\tilde{\Theta}$. Furthermore, one can obtain by $\tilde{\Theta} = \Theta - \hat{\Theta}$ that $\hat{\Theta}$ is bounded in probability.

(ii) Next, we prove finite-time convergence of $x$.

With (30) and the fact that $W$ is positive definite and radially unbounded with respect to $x$ and $\tilde{\Theta}$ in mind, it is directly deduced that

$$
\mathcal{L}W(x) \leq -\gamma W(x)^{\frac{4}{4-\omega}} - m_1 W(x) + (\Theta - \hat{\Theta})\dot{\tilde{\Theta}},
$$

$(33)$

where $\dot{\tilde{\Theta}} = \sum_{i=1}^{k} \tau_i \xi_i^4$ and $\gamma$, $m_1$ are two positive constants, respectively.

If $\Theta \leq \hat{\Theta}(0)$ with $\dot{\tilde{\Theta}} = \sum_{i=1}^{k} \tau_i \xi_i^4 \geq 0$, one has $\mathcal{L}W(x) \leq -\gamma W(x)^{\frac{4}{4-\omega}} - m_1 W(x)$.

If $\Theta \geq \hat{\Theta}(0)$, suppose that that there exists a finite time $T_1 \geq 0$ such that $\Theta \leq \hat{\Theta}(t)$ for any $t \geq T_1$. This means that $\mathcal{L}W(x(t)) \leq -\gamma W(x(t))^{\frac{4}{4-\omega}} - m_1 W(x(t))$ for any $t \geq T_1$.

Otherwise, there is another finite time $T_2$ satisfying $\dot{\hat{\Theta}}(t) = 0$ and $\Theta \leq \hat{\Theta}(t)$, which leads to $\mathcal{L}W(x(t)) \leq -\gamma W(x(t))^{\frac{4}{4-\omega}} - m_1 W(x(t))$, for any $t \geq T_2$.

Combining the aforementioned two cases, we learn that, for any $\hat{\Theta}(0)$, there is a finite time $T_3$ such that $\mathcal{L}W(x(t)) \leq -\gamma W(x(t))^{\frac{4}{4-\omega}} - m_1 W(x(t))$ for any $t \geq T_3$.

Since $0 < \frac{4}{4-\omega} < 1$ and $W(x(t))$ is positive definite and radially unbounded with respect to $x$, it can be proven that the solution $x$ is fast finite-time stable in probability by Lemma 5. □

## 5. Simulation Example

Consider the uncertain stochastic nonlinear system

$$\begin{cases} dx_1 = (x_2^{\frac{7}{5}} + \theta \sin x_1)dt, \\ dx_2 = u^{\frac{1}{3}}dt + x_2^{\frac{7}{5}}d\omega, \end{cases} \tag{34}$$

where $\theta$ is an unknown parameter.

In the simulation, Assumption 1 is satisfied with $r_1 = \frac{1}{2}, r_2 = \frac{5}{22}, r_3 = \frac{3}{22}, \omega = -\frac{2}{11} \in (-\frac{5}{14}, 0), p_0 = 1, p_1 = \frac{7}{5}, p_2 = \frac{1}{3}, \kappa_1 = 0, \mu_{11} = \bar{\mu}_{22} = 0, \bar{\mu}_{21} = -\frac{3}{11}$ and the virtual controller is designed as

$$x_2^* = -\xi_1^{\frac{5}{22}}(1 + \gamma + \hat{\Theta} + m_1 \xi_1^{\frac{2}{11}})^{\frac{5}{7}},$$

and one can construct the controller in the form (29).

Take the initial condition $[x_1, x_2]^T = [1, -1]^T$, and $\hat{\Theta}(0) = 0, \gamma = 1$, and $m_1 = 1$. In addition, the comparison results between the conventional finite-time controller (FTC) and the proposed fast finite-time controller (fast FTC) are shown in Figures 1 and 2, where the blue line describes FTC and the red dashed line represents fast FTC. Later, Figure 3 shows the trajectories $\hat{\Theta}(t)$. In addition, Figures 4 and 5 show that the convergent time becomes larger with the decrease in $m_1$ for fixed $\gamma$, and the convergent time is monotonously decreasing with $\gamma$ for fixed $m_1$.

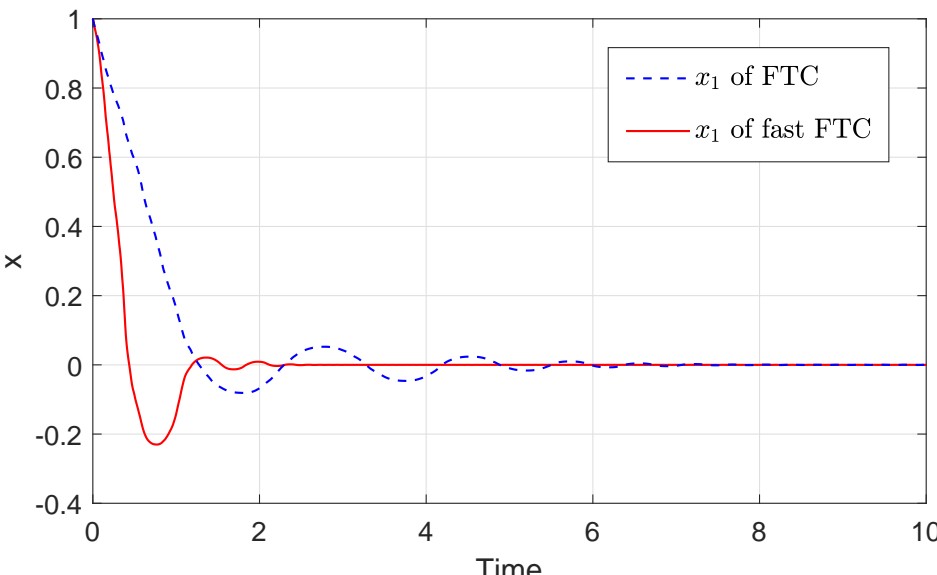

**Figure 1.** The trajectories of $x_1(t)$.

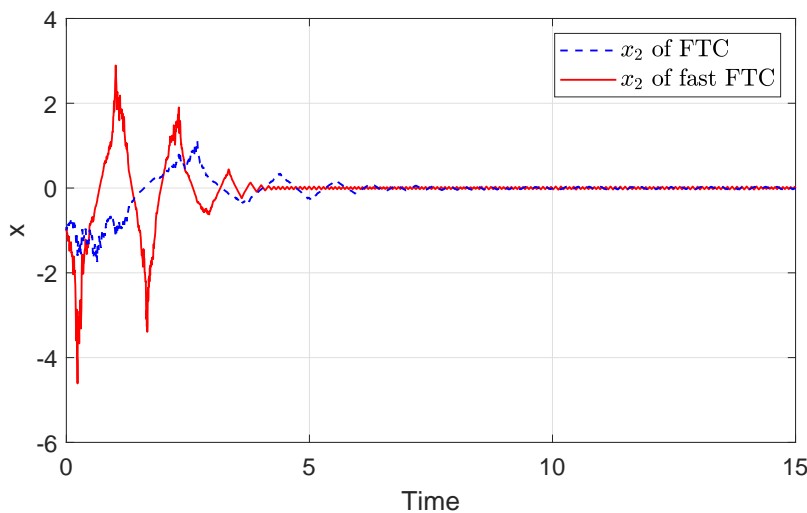

**Figure 2.** The trajectories of $x_2(t)$.

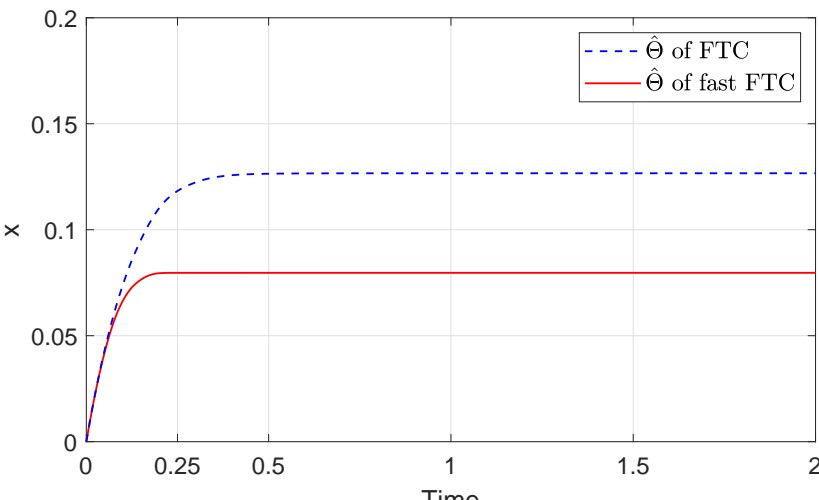

**Figure 3.** The trajectories of $\hat{\Theta}(t)$.

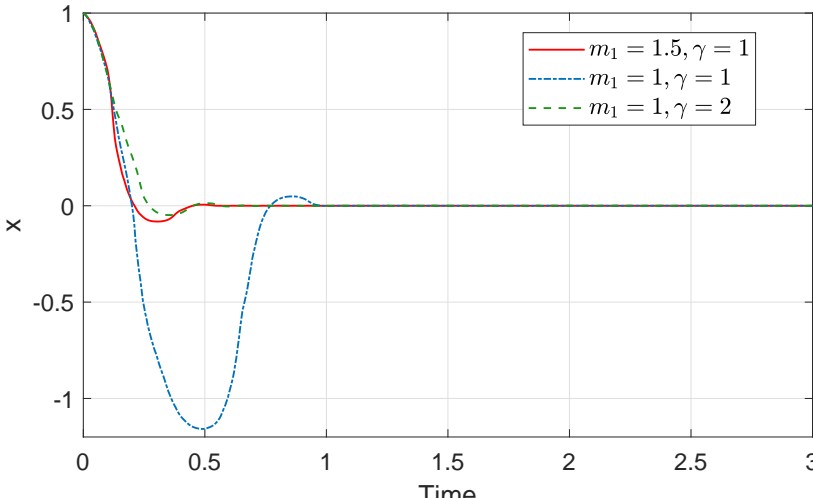

**Figure 4.** The trajectories of $x_1(t)$.

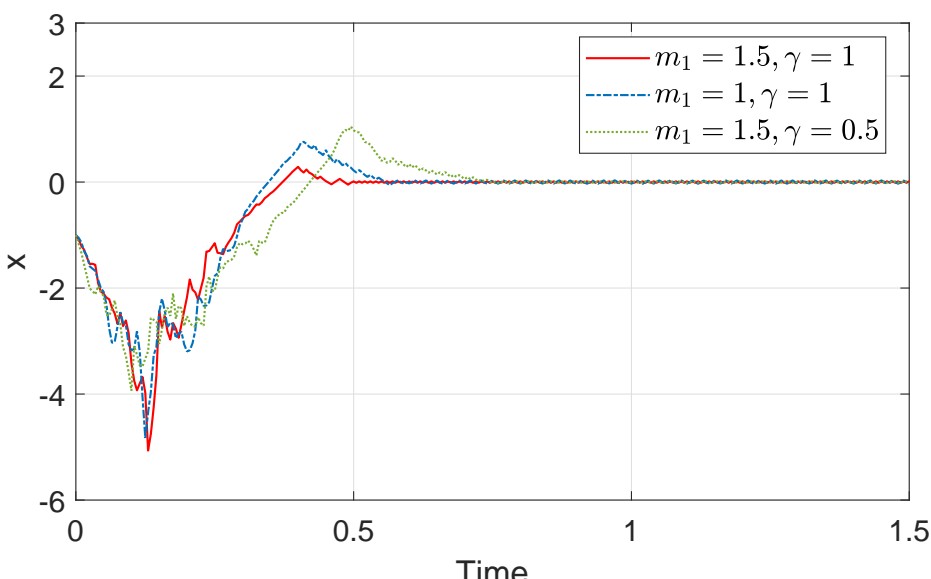

**Figure 5.** The trajectories of $x_2(t)$.

## 6. Conclusions

This paper has discussed an adaptive fast finite-time control for a class of HOSNSs. The first control obstacle lies in the fact that the stochastic nonlinear systems will generate more tedious inequalities since *Itô* stochastic differentiation involves not only gradients but also higher-order Hessian terms in the Lyapunov design process of HOSNSs. The second control obstacle comes from the fact that the adaptive and fast FTS of HOSNSs are the inherent obstacles caused by complex structures. To cope with this difficulty, a new adaptive fast finite-time control scheme is designed. Under the proposed scheme, the system's performance can be guaranteed in a finite time, and it also has better robustness and faster convergence speed.

**Author Contributions:** Conceptualization, Y.Y. and J.Z.; methodology, Y.Y. and J.Z.; software, Y.Y. and J.Z.; validation, Y.Y. and J.Z.; formal analysis, Y.Y. and J.Z.; writing—original draft preparation, Y.Y.; writing—review and editing, Y.Y.; supervision, J.Z.; project administration, J.Z.; funding acquisition, J.Z. All authors have read and agreed to the published version of the manuscript.

**Funding:** This work is supported by the National Natural Science Foundation of China under Grant 62173208, the Taishan Scholar Project of Shandong Province of China under Grant tsqn202103061, and the Shandong Qingchuang Science and Technology Program of Universities under Grant 2019KJN036.

**Institutional Review Board Statement:** Not applicable.

**Informed Consent Statement:** Not applicable.

**Data Availability Statement:** Not applicable.

**Conflicts of Interest:** The authors declare no conflict of interest.

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
