# Peer review of "Fast Finite-Time Stability and Its Application in Adaptive Control of High-Order Stochastic Nonlinear Systems"

_processes, doi:10.3390/pr10091676_

Round 1
Reviewer 1 Report
This article investigates the problem of adaptive fast finite-time control for a class of high-order stochastic nonlinear systems. In general, this paper is well written. The obtained results are original and interesting. The specific comments are addressed as follows:
1. The contribution of the paper needs to be further refined in the introduction. As far as journals are concerned, the innovation of articles is very important.
2. The authors should clarify the reasonability of Definition 1.
3. The mathematical derivations in this paper seem to be correct. The version needs to be checked again to avoid minor errors.
4. The format of references should be consistent, especially the authors, the punctuation and the journals in the references.
5. The literature review seems insufficient. Some latest results on stabilization of stochastic systems are necessary to recall in this paper, e.g., Automatica, vol. 136, 2022, Art. no. 110085; Automatica, vol. 125, 2021, Art. no. 109418. Automatica, vol. 143, 2022, Art. no.110471; SCIENCE CHINA Information Sciences, vol. 64, 2021, Art. No.200203; IEEE Transactions on Automatic Control, 2020, 66(12):5360-5367.They are highly related to the issue considered in this paper.
Author Response
Thanks you very much for recognition of our work. In what follows, we have
done our best to explain each comment seriously.
1. The contribution of the paper needs to be further refined in the introduction. As far as journals are concerned, the innovation of articles is very important.
Authors’ answer: First of all, we would like to express our sincere thanks for your detailed suggestions. In fact, we have made some remarks to elaborate on some of the innovations in this paper. Through careful deliberation, we have rewritten the language expression of the article. Please see page 2 for detail.
2. The authors should clarify the reasonability of Definition 1.
Authors’ answer: Thank you very much for your comments. In truth, the reasonability of definition of finite time stability in probability has been elaborated in the existing literature. Since a detailed elaboration of the definition is not the main innovation of this paper, we have simplified this section appropriately.
3. The mathematical derivations in this paper seem to be correct. The version needs to be checked again to avoid minor errors.
Authors’ answer: Thank you very much for your comments. After rechecking, we found some minor errors and corrected carefully them in the revised manuscript.
4. The format of references should be consistent, especially the authors, the punctuation and the journals in the references.
Authors’ answer: Thank you very much for your suggestion. The format of the References section has been modified according to the requirement of the journal.
5.The literature review seems insufficient.
Some latest results on stabilization of stochastic systems are necessary to recall in this paper, e.g., Automatica, vol. 136, 2022, Art. no. 110085;Automatica, vol.125,2021, Art.no.109418.Automatica, vol.143, 2022, Art.no.110471;
SCIENCE CHINA Information Sciences, vol. 64, 2021, Art. No. 200203; IEEE Transactions onAutomatic Control, 2020, 66(12): 5360-5367. They are highly related to the issue considered in thispaper.
Authors’ answer: Thank you very much for your suggestion. These articles are very useful toenhance the background of the introduction, we have added them into our references in the revised version.
Reviewer 2 Report
The finite time control is proposed for high-order stochastic nonlinear systems in this paper.
There exist some detail should be improved:
1. Give more detail about system (1), such as high order parameter p1,p2,…….
2. Some references should be check.
Author Response
The finite time control is proposed for high-order stochastic nonlinear systems in this paper. There exist some detail should be improved:
Authors’ answer: The authors would like to thank the reviewer very much for his suggestions, all of which lead to significant improvement of the quality of the manuscript. In what follows, we carefully respond to Reviewer’s comments.
1. Give more detail about system (1), such as high order parameter p1, p2,.
Authors’ answer: We fully agree with your suggestion and have changed this omission in the new version. This change makes the article more rigorous and is conducive to readers’ reading comprehension. Specifically, define pi to be a constant satisfying pi ∈ R≥1odd. Please see page 2 for detail.
2. Some references should be check.
Authors’ answer: Thank you very much for your careful comments, which is of great help to improve the quality of the article. In the revised manuscript, the format of the References section has been modified according to the requirement of the journal.

Reviewer 3 Report
The work presented by the authors is interesting; it is well supported by the cited bibliography and may be of interest to the industrial sector.
It is necessary to correct references 39 to 46 because they are incomplete and are not used in the work.
Reference 38 has not been used.
At the end of lemma 5 the variable that is bounded is not defined, its meaning can be inferred but it is not explicit in the text.
It is requested to incorporate a workflow of the proposal to facilitate the reading of the work and help the comprehension of the new proposal.
The example presented in section 5 is conclusive; it is worth detailing a little more some characteristics of the implementation process of the proposal of the work to find the solution of the controller.
It would be enriching that the authors incorporate a discussion about how to validate the proposal in more general cases in the conclusions section, since they are proposing a stabilization technique in nonlinear stochastic systems of high order
Author Response
Thanks you very much for recognition of our work. In what follows, we have
done our best to explain each comment seriously.
1. It is necessary to correct references 39 to 46 because they are incomplete and are not used in the work.
Authors’ answer: Thank you very much for your careful comments, which is of great help to improve the quality of the article. In response to the issue you has raised, we have made changes in the new version of the article.
2. Reference 38 has not been used.
Authors’ answer: Thank you very much for your careful comments, which is of great help to improve the quality of the article. In response to the issue you has raised, we have made changes in the new version of the article.
3. At the end of lemma 5 the variable that is bounded is not defined, its meaning can be inferred but it is not explicit in the text. It is requested to incorporate a work flow of the proposal to facilitate the reading of the work and help the comprehension of the new proposal.
Authors’ answer: We agree with the reviewers that bounded variables need to be defined in detail. For the sake of brevity in this paper, we cite the definitions in reference [29]. Therefore, we do not provide the relevant content of bounded variables in this paper. Please see page 4 for detail.
4.The example presented in section 5 is conclusive; it is worth detailing a little more some characteristics of the implementation process of the proposal of the work to find the solution of the controller.
Authors’ answer: We agree with the reviewers that the controller does require careful consideration and elaboration. For the sake of brevity in this paper, we have not elaborated the implementation process in detail in Section 5 Simulation. However, in Section 3 of the paper, we have provided a detailed proof of the proposed scheme. Simulation results have also shown that the system reaches
stability in a limited time.
5.It would be enriching that the authors incorporate a discussion about how to validate the proposal in more general cases in the conclusions section, since they are proposing a stabilization technique in nonlinear stochastic systems of high order.
Authors’ answer: Thanks you very much for recognition of our work. Your sincere suggestions and valuable comments are incorporated into the revised manuscript to enhance the quality of this paper. Finally, we faithfully hope that the revision addresses all the comments adequately, and are looking forward to hearing from you.

Reviewer 4 Report
The paper technically good written and application oriented but it suffers from the practical aspects and validation such as:
1) Design validation and demonstration have not been shown. At least compare two previously published work with your current results.
2) Real example can be considered to demonstrate the proposed work.
3) The little attention is given on simulation results. Show the robustness of the proposed approach by Monte Carlo Simulation by varying the system parameters.
4) The time history of the model in different phase with different applied control settings can be shown for better vision and validation of your model.
5) Sometimes the gain obtained in control theory will control the motion of object but in reality, gain may or may NOT be reality, therefore, put the constraint on gain and repeat the simulation.
The said comments can be addressed to improve the quality of the paper. The paper can be accepted after the major revision.
Author Response
Thank you very much for your careful comments. In what follows, we have
done our best to respond each comment seriously.
1. Design validation and demonstration have not been shown. At least compare two previously published work with your current results.
Authors’ answer: Thank you very much for your careful comments. We answer the reviewers comments from two aspects.
1. Different from the study of finite-time stable in probability in [28], the presented control manner is based on a new statement LV ≤ −γV α − m1V with 0 < α < 1 and γ, m1 > 0. However, the traditional finite-time control is motivated by the stability condition LV ≤ −γV α.
2. To demonstrate the validity of the proposed results, we show the comparison with the conventional finite-time controller in the simulation. With the equal parameters of the controller, the simulation results show that the scheme is feasible in practical applications.
2. Real example can be considered to demonstrate the proposed work.
Authors’ answer: Thank you very much for your careful comments. There is no doubt that the method proposed in this article is completely feasible in actual systems. For instance, for aircraftwing rock and parallel active suspension systems in the practical engineering systems, it is generally hoped that it can meet the desired control performance in fast finite-time.
3. The little attention is given on simulation results. Show the robustness of the proposed approach by Monte Carlo Simulation by varying the system parameters.
Authors’ answer: Thank you very much for your careful comments, which is of great help to improve the quality of the article. Following your comment, we revise the simulation parts to improve the presentation quality. Please see pages 9-10 for detail.
4. The time history of the model in different phase with different applied control settings can be shown for better vision and validation of your model.
Authors’ answer: Thank you for your suggestion. In the simulation of this paper, the control method in this paper and the finite-time control strategy proposed by Yin are used to design the controller respectively, as shown in Figure 1-2. The simulation results show that the scheme not only greatly accelerates the control speed, but also effectively shorten the settling time. In addition, we show the trajectory of the system state under different parameters by changing the parameters of the system, as shown in Figs. 4-5. We faithfully hope that the responses are satisfactory and adequately address your comments.
5. Sometimes the gain obtained in control theory will control the motion of object but in reality, gain may or may NOT be reality, therefore, put the constraint on gain and repeat the simulation.
Authors’ answer: First of all, we would like to express our sincere thanks for your detailed suggestions. Our research focuses on whether adding random items to the fast finite time study can achieve stability, which does not consider the influence of the gain.
6. The said comments can be addressed to improve the quality of the paper. The paper can be accepted after the major revision.
Authors’ answer: We tried our best to improve the manuscript and the changes have been marked in red in revision paper. We appreciate for Reviewers’s warm work earnestly, and hope that the correction will meet with approval. Once again, thank you very much for your comments and suggestions.

Round 2
Reviewer 4 Report
Most of the comments are addressed by the Authors. It can be accepted in the present form.